# Interfacial Enhancement by CNTs Grafting towards High-Performance Mechanical Properties of Carbon Fiber-Reinforced Epoxy Composites

**DOI:** 10.3390/ma16103825

**Published:** 2023-05-18

**Authors:** Wei Zhang, Mingfeng Dai, Xin Liang, Xi Wang, Wei Wei, Zuowan Zhou

**Affiliations:** 1School of Chemistry, Southwest Jiaotong University, Chengdu 610031, China; zw19102663364@163.com (W.Z.);; 2Key Laboratory of Advanced Technologies of Materials (Ministry of Education), School of Materials Science and Engineering, Southwest Jiaotong University, Chengdu 610031, China

**Keywords:** carbon fiber, coupling agent, CNTs, epoxy composites

## Abstract

The problem of interfacial interaction between carbon fiber (CF) and the matrix is the key to the failure of CF-reinforced plastic (CFRP). A general strategy to enhance interfacial connections is to create covalent bonds between the components, but this usually reduces the toughness of the composite material, which in turn limits the range of applications of the composite. In this study, carbon nanotubes (CNTs) were grafted onto the CF surface using the molecular layer bridging effect of the dual coupling agent to prepare multi-scale reinforcements, which significantly improved the roughness and chemical activity of the CF surface. By introducing a transition layer structure between the carbon fibers and the epoxy resin matrix to moderate the large modulus and scale differences between them, the interfacial interaction was improved while enhancing the strength and toughness of CFRP. We used amine-cured bisphenol A-based epoxy resin (E44) as the matrix resin and prepared the composites by the hand-paste method and performed tensile tests on the prepared composites, which showed that, compared with the original CF-reinforced composites, the modified composites showed an increase in tensile strength, Young’s modulus and elongation at break by 40.5%, 66.3% and 41.9%, respectively.

## 1. Introduction

Carbon fibers (CFs) have excellent properties such as high specific strength, high specific modulus, low coefficient of thermal expansion and corrosion resistance [1,2]. Therefore, they are commonly considered as an ideal material for reinforcing composites, which are widely used in various fields such as aviation aircraft, space rockets, traffic vehicles, wind power blades, etc. Although CF materials have excellent properties as a reinforcement, there are some key problems, such as the large modulus difference between the CF and the epoxy, which can easily lead to stress concentration during the stress process, resulting in poor mechanical properties of the composite material [3]. Moreover, the surface of CF is usually inert, making it difficult to bond well with the resin matrix [4]. As a unique component of composites, the strength of the interfacial interaction directly depends on whether the load can be uniformly and effectively transferred and dispersed between the matrix and the reinforcement, which further affects the mechanical properties of the material. Therefore, in order to improve the interfacial properties of composites, it is necessary to optimize the design of the interface [5]. Modifications of both the reinforcement and the matrix can improve the interfacial bonding. However, the modification of the reinforcement (fiber) is relatively simple and more easily controlled. Therefore, for CF-reinforced materials, the main optimization design is carried out on the CF surface [6].

In recent decades, a series of CF surface modification techniques have been developed, such as surface etching, surface activation [1,2], surface grafting [3,4,5,6,7,8] and surface coating [9,10,11,12]. Silane coupling agent grafting is an effective method for surface treatment of CF. The coupling agent has the feature of bifunctional groups [13,14,15], which can form a chemical bond with both CF and resin matrix. Grafting silane coupling agent on the surface of CF has two advantages, firstly, it can effectively increase the surface activity of CF, and secondly, it can significantly improve the infiltration ability of CF and matrix resin. In addition, it can act as a chemical bridge between the fiber and the resin matrix, thereby effectively improving the interfacial bonding properties of the fiber/resin matrix composite. However, it is difficult for a single coupling agent to have a good modification effect on the surface of CF.

In recent years, more researchers have focused on the assembly of carbon nanomaterials on CF, and the commonly used methods include nanoparticle modification [16,17,18], graphene surface modification [19,20,21] and CNTs modification [22,23], which is especially worth investigating. The introduction of nanomaterials can bring their excellent properties to composites [24,25], such as wave absorption, thermal insulation, sensing [26] and mechanical strength. However, for engineering applications the mechanical strength of structural components is often the focus of our attention. Among the many carbon nanomaterials, CNTs, SiO_2_ and GO are commonly used to prepare fiber-reinforced polymer composites, among which the structural properties of CNTs with high aspect ratios give them high strength and toughness, so that they can exhibit excellent mechanical properties and thus can be used as fillers in resin-based composites or as coatings on fiber surfaces to improve the strength of the composites. Compared with CNTs alone, surface functionalization modification of CNTs by introducing small molecules can significantly improve the dispersion [27] and compatibility [28] of CNTs in composites, thus enhancing the composite properties. Yao [17] deposited carboxyl-functionalized carbon nanotubes on the surface of carbon fibers. The interlayer shear strength (ILSS) of CNTs/CF/epoxy composites increased by 58.6%. However, it is worth noting that if CNTs are only “physically adsorbed” on the fiber surface, when the interface is subjected to external forces, it can easily separate from the fiber surface and interfacial debonding occurs and affects the composite properties. Therefore, a more efficient method to immobilize CNTs is needed. The preparation of CNTs/CF multiscale reinforcements by chemical grafting of nanoscale CNTs to microscale CF surfaces can effectively solve this problem. Xu [29] used polydopamine (PDA) to introduce CNTs into the surface of CF through π–π interactions and covalent bonding, which resulted in an 89.72% increase in the interfacial shear strength (IFSS) of the modified CF/epoxy composites. The IFSS is suggested to be dependent on the shape of the fiber. Typically, in perfect adhesion, the IFSS increases non-linearly from the center of the fiber to a peak value at the end. This is assuming a uniform cylindrical fiber shape during elastic load transfer [30]. However, fibers such as collagen fibrils of the extracellular matrix of connective tissues [30] and nanoparticles such as hydroxyapatite [31] have symmetric tapered ends. During elastic loading, the unique shape of these fibers and particles results in the generation of interfacial shear stresses which are nearly uniformly distributed along the fiber axis. The uniform interfacial shear stress profile serves to minimize the yielding of interfacial bonds. This could otherwise reduce the effectiveness of stress transfer from the matrix to the fiber (or particle) during elastic loading [30] and during fiber pull-out [32]. Carbon fibers do not have tapered ends, which makes generating uniform interfacial shear stress along the carbon fiber axis unfeasible. Therefore, modifying the carbon fiber surface using the adhesion argument is the best way to address the issue of effective stress transfer. However, these strategies were mainly aimed at improving the interfacial bond strength, which usually degrades the toughness of the composites and limits their range of applications. Therefore, there is an urgent need for a modification method that can have the characteristics of strengthening and toughness at the same time.

In this study, we prepared a multi-scale reinforcement (CNTs-4-KH560-KH792-6-CF) by bridging CNTs and carbon fibers using a dual coupling agent molecular layer. This multi-scale construction method can create a transition layer between epoxy resin and CF, which can ease the modulus and scale difference between the fiber and resin and thus improve the interfacial bonding properties and simultaneously the mechanical properties and toughness of the composite. We investigated the effects of surface-grafted CNTs on the surface morphology as well as the chemical structure of CF and discuss in detail the performance enhancement mechanism of the modified carbon fiber-reinforced composites.

## 2. Experimental Methods

### 2.1. Materials

Carbon fiber fabric (CFF, PAN-based, Chengdu Luchen New Material Technology Co., Chengdu, China), carboxylated multi-walled carbon nanotubes (inner diameter 5–12 mm, outer diameter 30–50 nm, length 10–20 μm, COOH ≥ 2%) (≥ 95%), N-(2-aminoethyl)-3-aminopropyltrimethoxysilane (KH792) (AR), γ-(2,3-epoxypropoxy)propytrimethoxysilane (KH560) (AR) were provided by Shanghai Aladdin Biochemical Technology Co. (Shanghai, China). Epoxy resin E44 was provided by Shandong Urso Chemical Technology Co. (Jinan, China). Curing agent diethyltoluenediamine was provided by Jinan Noser New Material Co. (Jinan, China). Anhydrous ethanol (≥ 95%) was provided by Kolon Drugs. Deionized (Chengdu, China) water was homemade by the laboratory.

### 2.2. Preparation of CNTs/CF Multi-Scale Reinforced Composites

First, surface modification of CFF with KH792 solution (the solvent was composed of ethanol and deionized water mixed at a 95:5 volume ratio) at concentrations 6 wt% was carried out for 2 h at 120 °C, and the resulting amidated CFF was named KH792-6-CF. Secondly, CNTs were added into the mixed solvent (ethanol and deionized water 95:5 by volume), sonicated (200 W) and dispersed for 1 h. A certain amount of KH560 was added to prepare a 4 wt% KH560 solution and the grafting reaction was carried out with mechanical stirring for 4 h. The epoxy-functionalization of CNTs was completed, and the resulting epoxy-functionalized CNTs were named CNTs-4-KH560, and the suspension of CNTs-4-KH560 was obtained at the same time. Third, KH792-6-CF was submerged in the CNTs-4-KH560 suspension for 5 min to absorb the KH560-modified CNTs, followed by the drying process in an oven at 120 °C for 0.5 h. This process was performed twice to elevate the absorption amount. The obtained samples were named CNTs-4-KH560-KH792-6-CF. The preparation mechanism diagram is shown in Figure 1. Finally, the epoxy resin–hardener mixture was applied to the CFF using a hand-paste molding process, and the CFF was laid in alternating and vertical stacks, and the excess resin was scraped off with a tool after laying one layer of CFF, for a total of four layers, which were cured at 120 °C and 170 °C for 2 h each. The obtained composite was named carbon fiber-reinforced composite (CFRP). Then, the test strips were prepared by a diamond cutter with reference to GB/T1040.2-2006 (5A), and five specimens were prepared for each composite material.

### 2.3. Material Characterization

Transmission electron microscopy (TEM) characterization of CNTs was performed on a JEM-2100F (JEOL, Japan Electronics Co., Tokyo, Japan). Scanning electron microscopy (SEM, JEOL, JSM-IT500, Japan Electronics Co., Tokyo, Japan) was used to observe the morphology of CF and its complexes. Fourier transform infrared spectrometry (FTIR, INVENIOR, Bruker, Mannheim, Germany) was used to analyze the functional group changes during the process. X-ray photoelectron spectroscopy (XPS) tests were performed on a Thermo Fisher ESCALAB Xi+ spectrometer (Thermo Fisher ESCALAB Xi+, Thermo Fisher, Waltham, MA, USA) for the analysis of chemical composition changes on the CF surface. The wettability of CF before and after modification was characterized using a contact angle measuring instrument. Five specimens were tested for each group of samples, and the probability value of *p* < 0.05 was used as the criterion for significant differences. The calculated means and standard deviations were also expressed as bar graphs. The dispersive and polar components can be calculated by the following Equations (1) and (2):(1)$$\gamma_{l}\left(1+cos\theta \right)=2{\left(\gamma_{l}^{d}\gamma_{s}^{d}\right)}^{1/2}+2{\left(\gamma_{l}^{p}\gamma_{s}^{p}\right)}^{1/2}$$
(2)$$\gamma_{s}=\gamma_{s}^{d}+\gamma_{s}^{p}$$
in which $\gamma_{l}$, $\gamma_{l}^{d}$ and $\gamma_{l}^{p}$ are the surface tension, dispersive and polar component of immersion liquid, respectively. $\gamma_{s}$, $\gamma_{s}^{d}$ and $\gamma_{s}^{p}$ are the surface energy, dispersion component and polarity component of carbon fiber, respectively.

The mechanical properties of the composites were tested using a material universal testing machine (UTM4304X, Shenzhen New Sansi Materials Testing Co. Ltd., Shenzhen, China). The mechanical properties of composites were tested in accordance with GB/T1040.2-2006 (5A), and the dimensions of the sample strips were also in accordance with this standard. Five specimens were tested for each group of samples, and the tensile rate in the test was 1 mm/min. The probability value of *p* < 0.05 was used as the criterion for significant differences. The calculated means and standard deviations were also expressed as bar graphs.

## 3. Results and Discussion

By bridging CNTs and CF using dual coupling agent molecular layers, CNTs can be better and more uniformly grafted to the CF surface, establishing a multi-scale reinforcement with synergistic molecular, nano- and microscale components. We investigated the pattern of CNTs grafting on the physicochemical structure change on the CF surface through a series of characterization studies, and analyzed and elaborated the mechanism of CNTs grafting on the mechanical properties of CF-reinforced composites.

### 3.1. Characteristics of Modified CNTs

#### 3.1.1. Surface Morphology Analysis

In order to investigate the structural changes on the surface of the coupling agent graft-modified CNTs, TEM tests were performed on the modified CNTs, and the test results are shown in Figure 2. Compared with the untreated CNTs, the surface of the coupling agent-modified CNTs formed an obvious layered coating of amorphous carbon, which was found to be caused by the formation of modified layers after successful grafting of the coupling agent (Appendix A).

#### 3.1.2. Surface Chemical Analysis

In order to investigate the chemical structure changes in the coupling agent graft-modified CNTs and to determine the composition of the surface modified coating, FTIR tests were performed on the modified CNTs, and the results are shown in Figure 3. The characteristic C-H, C=O and C=C peaks of CNTs were observed at 2920/2851 cm^−1^, 1734 cm^−1^ and 1634 cm^−1^ (Figure 3a), respectively. Compared with the untreated CNTs, the coupling agent-treated FTIR spectra showed peak abrupt changes near 1271 cm^−1^, 1087 cm^−1^, 1049 cm^−1^ and 879 cm^−1^ for the epoxy group [33], as Si-O-C, C-O and Si-O-Si characteristic absorption peaks (Figure 3b), respectively, indicating that the addition of the coupling agent generated new chemical bonds on the surface of the CNTs, resulting in peak mutation, confirming that the coupling agent was successfully grafted on the surface of CNTs.

### 3.2. Characteristics of Modified CF Surfaces

#### 3.2.1. Surface Morphology Analysis

SEM was used to describe the morphological changes in CF before and after graft modification by introducing CNTs, and the results are shown in Figure 4. The pristine CF shows a smooth surface (Figure 4a,d), and when the CF was treated with KH792 modification, the obtained modified CF appeared to have a coating on the surface (Figure 4b,e), and there is a certain increase in roughness. Further multi-scale reinforcements were prepared by introducing KH560-modified CNTs onto the KH792-modified CF through chemical bonding. It can be seen that this modification method enables CNTs to have a high grafting density on the CF surface and even form a graft-modified layer with a certain thickness, which also leads to a sharp increase in the roughness of the CF surface and improves the anchoring sites for mechanical interlocking with the substrate. At the same time, this modification means that a large number of gaps and pores remain on the surface of the CF, which enables the resin matrix to better infiltrate into the fiber surface.

#### 3.2.2. Surface Chemical Analysis

FTIR tests were performed on CFs to investigate the effect of modification on their surface chemistry. There are new peaks on the CF infrared spectra after the coupling agent treatment. The KH792-6-CF can be compared with the pristine CF in Figure 5b where some new absorption peaks were generated at 1521 cm^−1^, 1264 cm^−1^, 1123 cm^−1^ and 878 cm^−1^ for the characteristic groups N-H, Si-C, Si-O-C and Si-O-Si, respectively [13], indicating that the addition of the coupling agent has created new chemical bonds within the composites. Further FTIR results of CNTs-4-KH560-KH792-6-CF showed that the peak at 1521 cm^−1^ disappeared, indicating that the amine group reaction on the CF surface was complete. Furthermore, the appearance of new characteristic peaks at 1271 cm^−1^, which were analyzed as characteristic absorption peaks of epoxy functional groups [33], indicates that there were epoxy functional groups on the outer surface of the CNTs that were not involved in the grafting process after the grafting was completed. These remaining epoxy groups can enhance the wettability of carbon fibers with epoxy resin matrix.

The chemical composition of the sample surface was investigated by XPS. The wide-scan XPS results are shown in Figure 6a, where the CF surface is mainly composed of carbon (284.7 eV) and oxygen (532.5 eV). After modification, the nitrogen (399.8 eV) and silicon (153 eV, 101 eV) increased significantly. The content of N and Si elements in KH792-6-CF increased from 0 to 5.47% and 6.11%, respectively, compared with the original CF, while the O/C ratio also increased to some extent. This is due to the introduction of new functional groups in the coupling agent, which confirms the successful grafting of the coupling agent on the CF surface. Meanwhile, compared to KH792-6-CF, the content of N element on the surface of CNTs-4-KH560-KH792-6-CF decreased (from 5.47% to 1.85%), while the increase in Si element content and O/C ratio was more obvious, which was the result of the introduction of functional groups (C-Si, epoxy groups and O-C=O) to the CF surface during the grafting process of epoxy-functionalized CNTs.

In order to better study the effect of modification treatment on the chemical structure of the CF surface, the C1s peaks in the XPS spectra were split and the results are shown in Figure 6. The high-resolution XPS spectrum of C1s pristine CF (Figure 6b) consists of four peaks [24,34] corresponding to C=C (284.6 eV), C-C (285.4 eV), C-O (286.4 eV) and O-C=O (289 eV). Compared with CF, KH792-6-CF showed some new carbon peaks at 286.1 eV and 283.9 eV (Figure 6c) that are attributed to C-N and C-Si in KH792 [14,15]. It should be noted that compared to pristine CF, KH792-6-CF showed a very significant increase in the area of C-N (286.1 eV) and C-Si (283.9 eV) (Appendix A), and all the remaining groups decreased to varying degrees. Since XPS is a surface element analysis technique, it can be concluded that the grafting of the coupling agent forms a homogeneous coating on the fiber surface, which further indicates the successful grafting of the coupling agent. There is no new peak generation for CNTs-4-KH560-KH792-6-CF compared to KH792-6-CF. However, it can be observed that the area of C-O (286.4 eV), O-C=O (289 eV), C-Si (283.9 eV) has increased to some extent, and for the area of C=C (284.6 eV), C-C (285.4 eV) and C-N (286.1 eV) a decrease occurred (Appendix A). The principle is also similar to the masking effect of the epoxy-functionalized CNTs coating on the KH792-6-CF surface mentioned earlier, exposing more characteristic functional groups on the epoxy-functionalized CNTs surface (C-O, O-C=O, C-Si).

The changes in surface chemical composition and microstructure can affect fiber surface energy, and a higher surface energy can lead to better wettability and stronger interfacial adhesion between fiber and matrix. Figure 7 shows the contact angle (*θ*), surface energy (*γ*), dispersion component (*γ^d^*) and polar component (*γ^p^*) of different CF surfaces. The size of the polar component is related to the content of polar groups on the surface of the carbon fiber, and the size of the dispersion component is related to the specific surface area and roughness of the CF, which together affect the surface energy of the fiber. Pristine CF with a non-polar and smooth graphite surface showed the lowest value of surface energy (32.11 mN/m). After grafting KH792 onto the surface of CF (KH792-6-CF), there was an obvious decreasing tendency in the contact angle and an increasing tendency in the surface energy (42.59 mN/m). The increase in the polar component of the surface energy can be interpreted from the increase in polar amine groups in KH792. Therefore, the wettability between the amino-functionalized CF and the polar polymer matrix is significantly improved. In addition, the increase in the dispersion component, due to the increase in roughness, also contributes to the increase in the surface energy. The surface of epoxy-functionalized CNTs contains a large number of polar groups, so the grafting of CNTs introduces a large number of polar groups to the surface of CF, which increases the polar fraction of the fiber surface. The magnitude of the dispersion fraction is related to the surface roughness of the carbon fiber, and it can be seen from the SEM test (Figure 4f) that CNTs grafting can increase the surface roughness of the CF, so its dispersion fraction has a substantial increase compared with KH792-6-CF. Compared with KH792-6-CF, the modified reinforcement (CNTs-4-KH560-KH792-6-CF) has higher surface energy (50.00 mN/m), which indicates that the grafting of CNTs on CF can effectively improve the roughness and wettability.

#### 3.2.3. Mechanical Properties of Composites

The mechanical properties of the CF-reinforced composites prepared by different CF modification routes show significant differences (Figure 8a). In the absence of a coupling agent, the tensile strength and elastic modulus of pristine CF-reinforced composites (CFRP) are only 519 MPa and 12 GPa. With the grafting of the coupling agent containing amine functional groups onto the carbon fiber surface, the surface energy of the carbon fiber increases and allows the matrix resin to better infiltrate the fiber surface while avoiding the appearance of interfacial defects during the preparation of the composite. When considering the issue of interfacial defects on a smaller length scale, at the level of the CNTs, it has been demonstrated that Stone–Wales defects can arise on CNTs [35]. This structural imperfection occurs when two pairs of hexagons are transformed into a pentagon–heptagon formation through Stone–Wales bond rotation. The strength of the nanotube is thought to be reliant on various factors, including the nanotube’s structure (diameter and chirality) and external conditions such as temperature and strain rate [35]. If structural defects are present on the outer shell of MWCNTs, the strength could be notably lower than in the absence of defects [35]. However, molecular mechanics predictions indicate that the defects could potentially enhance the mechanical integrity of the CNTs by way of defect–defect interactions [36]. It was discovered that at short interdefect distances, the interatomic bond energy is high, but as the distance between defects grows, the bond energy decreases and stabilizes at a consistent level [36]. High interatomic bond energies are advantageous because they signify that more energy is needed to separate the atoms in the defective region, which would otherwise have an energetically unstable conformation [36]. At the same time, the amidated CF is able to bond with the matrix resin in the form of stable chemical bonds and enhances the interfacial effect. Macroscopically, the tensile strength and tensile modulus of the coupling agent graft-modified CF-reinforced composite (KH792-6-CFRP) increased to 633 MPa and 17 GPa, respectively, which were 22.0% and 41.7% higher than those of CFRP. Further epoxy-functionalized CNTs were introduced to build a multi-scale structure on the CF surface that can better act as a buffer layer and disperse stresses, in addition to the ability of CNTs to enhance the interfacial interaction effect by physical anchor riveting action, which in turn improves the mechanical properties of the composites. Among them, the tensile strength and tensile modulus of CNTs graft-modified CF-reinforced composites (CNTs-4-KH560-KH792-6-CFRP) increased to 729 MPa and 21 GPa, respectively, which were 40.5% and 66.3% higher than those of CFRP (Appendix A). Additionally, the elongation at break increased from 3.9% to 5.6%, an improvement of 41.9%. In addition, the modified carbon fiber reinforced composites also showed good performance in the three-point bending test (Appendix A). It was confirmed that the interfacial interaction of the modified treated composites was enhanced.

The digital photographs of the composites after fracture are shown in Figure 9a–c. Compared with the unmodified carbon fiber-reinforced composites, which showed a large amount of fiber pull-out (Figure 9a), the fiber pull-out phenomenon at the fracture of the coupling agent alone and carbon nanotube graft-modified carbon fiber-reinforced composites was reduced (Figure 9b,c), with the latter showing a further reduction in the number of pulled-out fibers (Figure 9c), indicating a better interfacial effect. In order to further evaluate the enhancement effect of the modification on the interfacial interaction, the surface micromorphology of the fibers pulled out of the fracture surface was characterized. The surface micromorphology of the pulled-out fibers in the tensile fracture surfaces of CF-reinforced composites modified in different stages (referred to as CFRP, KH792-6-CFRP and CNTs-4-KH560-KH792-6-CFRP) is shown in Figure 9d–i. As shown in Figure 9g, the tensile fracture morphology of the original CF-reinforced composite exhibits the classic fiber pull-out appearance with almost no resin adhesion, which implies a weak interfacial interaction between the fibers and the epoxy resin. If the CF is modified with KH792, the situation is much better (Figure 9h), because the amino groups on the surface of the modified CF can improve the interfacial interaction effect by forming a stable chemical bond with the matrix. By further introducing CNTs-4-KH560 into the KH792-modified CF to prepare a multi-scale reinforcement (CNTs-4-KH560-KH792-6-CF), the grafted CNTs act as a physical anchor point to improve the interfacial interaction effect. We found a large amount of resin fragments on the surface of the extracted CFs and the resin@CNTs structure could be observed (Figure 9i). This indicates a prominent interfacial interaction between the resin matrix and the CNTs-4-KH560-KH792-6-CF. 

## 4. Conclusions

The surface modification of CF was completed by grafting nanoscale CNTs onto the microscale CF surface through a double coupling agent molecular layer as a bridging agent, and the CNTs-4-KH560-KH792-6-CF multi-scale reinforcement was successfully prepared. The results show that the CNTs grafted on the CF surface significantly increase the roughness of the CF surface, and CNTs can enhance the interfacial interaction effect through the physical anchor riveting effect. In addition, the successful grafting of CNTs also introduced a large number of polar functional groups to the CF surface, which changed the chemical environment of the CF surface and increased the wettability of the fiber surface. It can also bond better with the resin matrix through chemical bonding, thus enhancing the interfacial effect. The modification also introduces a transition layer structure between the CF and the resin matrix, which moderates the problem of excessive modulus and scale difference between the two, reduces stress concentration, enhances the interfacial effect and thus improves the mechanical properties. Compared with the original CF-reinforced composites, the prepared composites achieved simultaneous improvement in three aspects of mechanical properties, with the tensile strength, Young’s modulus and elongation at break increasing by 40.5%, 66.3% and 41.9%, respectively. The modification method described in this paper has some potential problems in terms of scale production, but the available results indicate that the scheme still has great potential for the preparation of high-strength and high-toughness carbon fiber composites.

## Figures and Tables

**Figure 1 materials-16-03825-f001:**
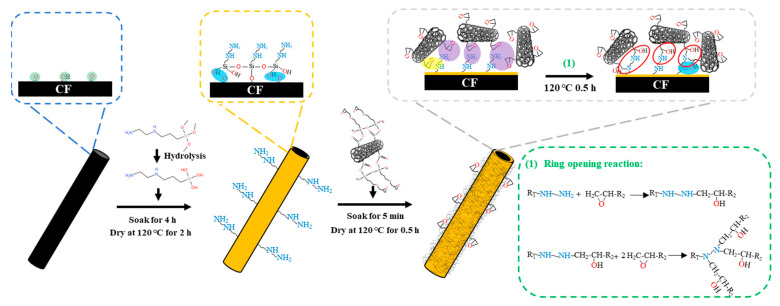
Schematic diagram of the preparation process: CNT graft-modified CF for the preparation of multi-scale reinforcement (CNTs-4-KH560-KH792-6-CF).

**Figure 2 materials-16-03825-f002:**
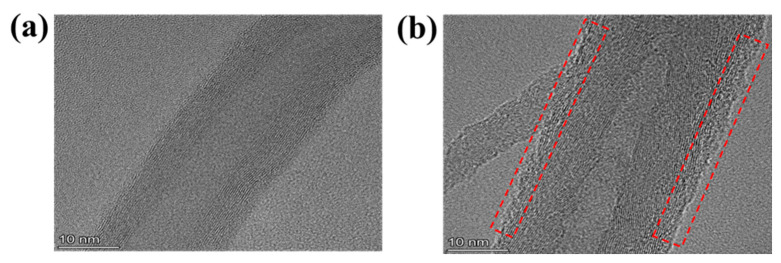
TEM images of modification CNTs: (**a**) untreated CNTs, (**b**) CNTs-4-KH560.

**Figure 3 materials-16-03825-f003:**
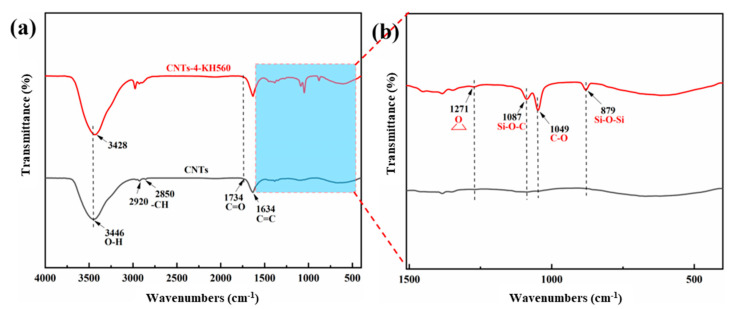
(**a**) FTIR spectra of modification CNTs, (**b**) enlarged absorption peaks in a range of 1510–400 cm^−1^ as marked in (**a**).

**Figure 4 materials-16-03825-f004:**
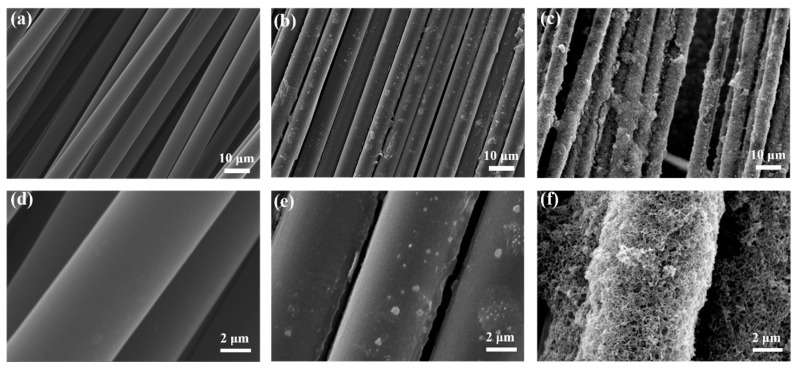
Microscopic morphology of: (**a**,**d**) pristine CF, (**b**,**e**) KH792-6-CF, (**c**,**f**) CNTs-4-KH560-KH792-6-CF.

**Figure 5 materials-16-03825-f005:**
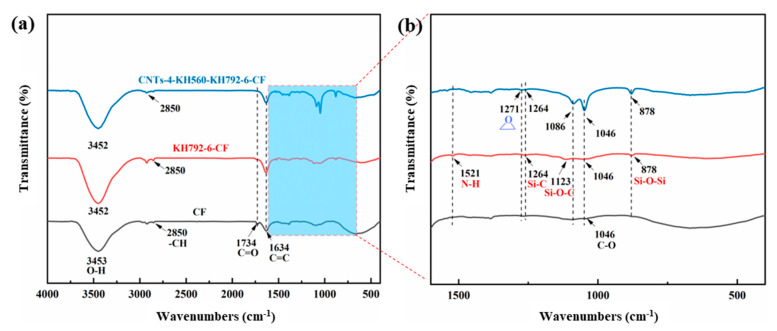
(**a**) FTIR spectra of different modified CF (**b**) enlarged absorption peaks in a range of 1607–400 cm^−1^ as marked in (**a**).

**Figure 6 materials-16-03825-f006:**
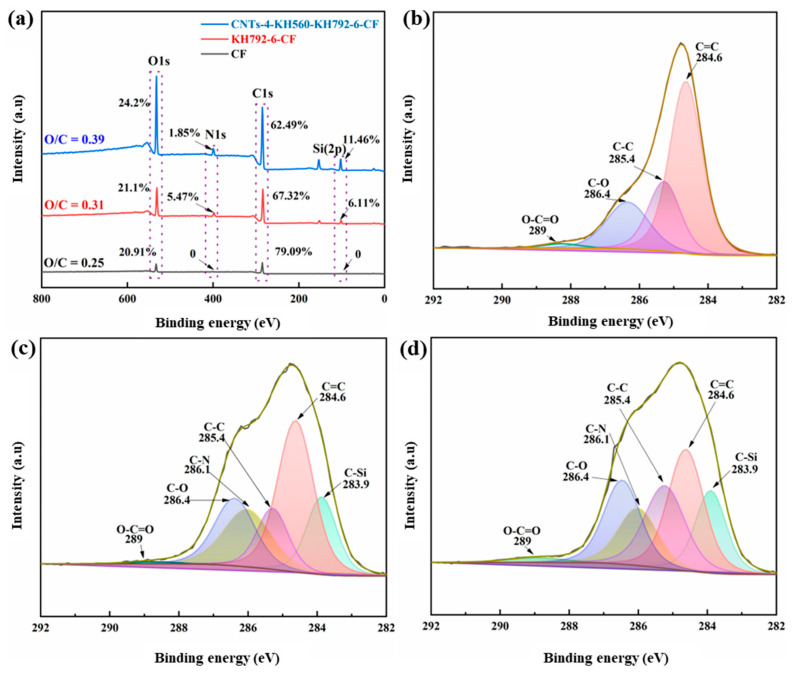
(**a**) XPS wide-scan measurement spectra, C1s high-resolution XPS spectra split peak fits: (**b**) pristine CF, (**c**) KH792-6-CF, (**d**) CNTs-4-KH560-KH792-6-CF.

**Figure 7 materials-16-03825-f007:**
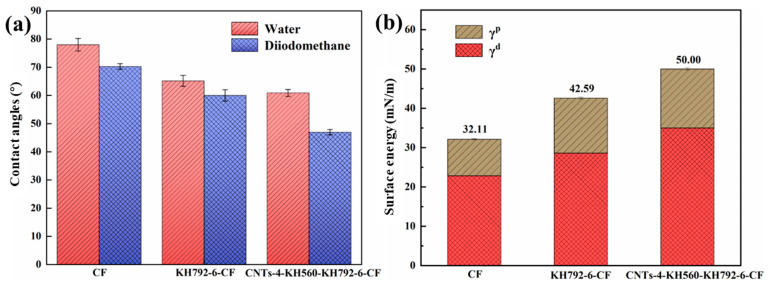
Wettability of CF surface: (**a**) contact angle; (**b**) surface energy.

**Figure 8 materials-16-03825-f008:**
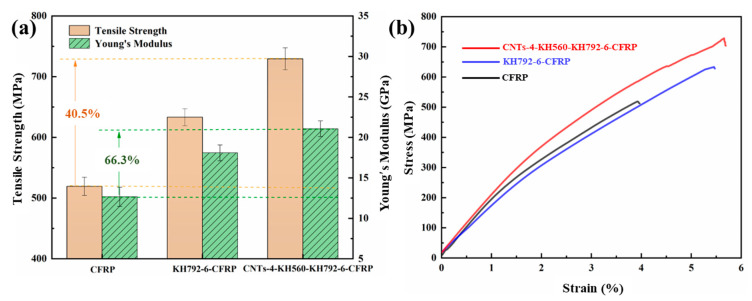
Mechanical properties of composites: (**a**) the results of tensile strength and Young’s modulus, (**b**) CF-reinforced composite tensile test stress–strain curve.

**Figure 9 materials-16-03825-f009:**
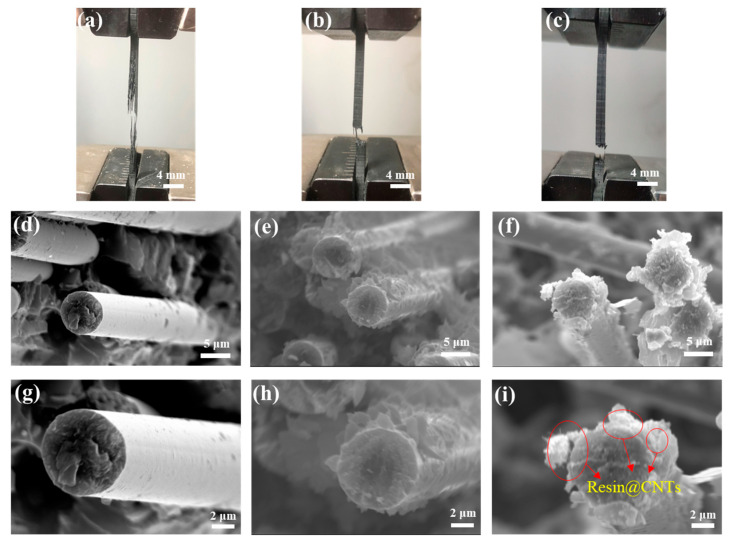
Digital photos of composite material after fracture: (**a**) CFRP, (**b**) KH792-6-CFRP, (**c**) CNTs-4-KH560-KH792-6-CFRP**;** morphology of the fracture surface of different modified reinforced composites: (**d**,**g**) CFRP, (**e**,**h**) KH792-6-CFRP, (**f**,**i**) CNTs-4-KH560-KH792-6-CFRP.

## Data Availability

Data will be made available on request.

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
