# Peer review of "Interfacial Enhancement by CNTs Grafting towards High-Performance Mechanical Properties of Carbon Fiber-Reinforced Epoxy Composites"

_materials, 2023, doi:10.3390/ma16103825_

Round 1

Reviewer 1 Report

In the paper "Interfacial enhancement by CNTs grafting towards high-performance mechanical properties of carbon fiber reinforced epoxy composites", the authors investigate the effect of CNTs grafting on the interfacial properties and mechanical properties of carbon fiber reinforced epoxy composites. The results show that CNTs grafting can significantly improve the interfacial bonding between the carbon fibers and epoxy matrix, leading to improved mechanical properties . The paper is interesting and can be accepted for publication after major revision. The following issues should be clarified. 

It is hard to understand which form have carbon fibers, in some places it is mentioned as fibers, in others as carbon fiber fabric. More extensive information should be added to this topic. The measurement of the wettability of the carbon fibers is quite a difficult task, and if I understood rightly, all studies were conducted on straightened carbon fiber fabric. The morphology of this object is likely wavy, so maybe it makes sense to use a more comprehensive analysis of this surface to take into account surface roughness.

Maybe it makes sense to calculate the amount of grafted CNTs per m2 of the carbon fiber fabric. 

While the paper reports improvements in tensile strength, Young's modulus, and elongation at break, it is unclear whether the modified composites exhibit improvements in other important properties, such as fatigue resistance or impact strength. Appropriate discussion should be provided.

The paper mentions that creating covalent bonds between the components can improve their mechanical properties, but it does not discuss potential limitations or drawbacks of the approach used in this study. A more detailed discussion of potential limitations, such as the scalability or cost-effectiveness of the method, would help readers understand the potential practical implications of the research. Additionally, it will be valuable to compare the effectiveness of this approach to other interfacial enhancement approaches. A comparison to other methods, such as fiber surface modification or matrix modification, would provide a more comprehensive understanding of the potential benefits and limitations of CNT grafting.

Finally, I suggest citing the papers where carbon nanotubes were modified:

https://doi.org/10.1016/j.apsadv.2021.100104

 https://doi.org/10.1002/anie.200353329

The quality of the English is well.

Reviewer 2 Report

The manuscript discusses grafting of CNTs on the surface of carbon fiber in order to enhance the interfacial strength of carbon fiber reinforced polymers based composites. Fiber matrix debonding is the weakest link and is normally one of the first micro-damage modes that occurs. Therefore, it is of high importance to strengthen this weakest link.  

I recommend to publish this manuscript with the following modifications/responses:

1.            Mention the materials system, fabrication method and type of tests that were conducted in the abstract.

2.            Refer the figure of pristine CF as Figure 2 (a) and (d) correctly in line 137. Also correct the spelling of shows.

3.            No details of the mechanical test, equipment, size of samples were found in the manuscript.  

4.            What ASTM standard was used for tensile test?

5.            Also show fractured samples in figure 7 in addition to SEM.

6.             I believe that fiber/matrix interface decohesion plays an important role in shear loading rather than in-plane tensile loading. As under in-plane tensile loading, fibers are mainly supporting the load instead of matrix or the interface. Therefore, additional shear tests should be added to better quantity the effects of CNTs grafting.

7.             Add discussion about the other carbon-based nanomaterials. Why CNTs were selected. Add the following articles to give insights into the applications of other nano-materials in line 58-59 for the developing of multifunctional composites.

https://doi.org/10.1016/j.coco.2022.101382

English is fine. Spell check is required.

Reviewer 3 Report

This manuscript reports a study on CNTs grafting on carbon fibers as a way of enhancing the mechanical properties of carbon fiber reinforced epoxy composites. Although there are several studies reported on this topic, the authors claimed that so far the techniques used did not lead to desirable outcome, e.g. the method of physical bonding of CNT to surface of CF is not robust, chemical grafting of CNTs to CF surface by PDA improved the composite toughness but not strength. So they introduced a method known as using the molecular layer bridging effect of the dual coupling agent to graft the CNT onto the CF. I think this is novel.

I have some comments and suggestions for the authors, for the purpose of improving this manuscript.

Question 1

With regard to Figure 5a, what do the error bars refer to? It looks like the values for each bar reflect the mean values. What mean values are these referring to? How many samples/times of measurement were taken? Please explain these in the manuscript.

Question 2

With regard to Figure 5b, how did the authors determine the surface energies? It looks like they were determined from Eq. (1) but how? Please add a description in the manuscript?

Question 3

With regard to figure 6a, what do the error bars refer to? It looks like the values for each bar reflect the mean values. What mean values are these referring to? How many samples of measurement were taken? Please explain these in the manuscript.

Question 4

With regard to Figure 6b, why was the stress not equal to zero at strain = 0, for the CNTs-4-KH560-KH792-6-CFRP specimen?

Question 5

To strengthen and balance the argument for enhancing the IFSS of fibres, that could benefit the readers of this journal, please add the following sentences to Line 70, after 'Xu [26] used polydopamine (PDA) to introduce CNTs into the surface of CF through π-π interaction and covalent bonding, which resulted in an 89.72% increase in the interfacial shear strength (IFSS) of the modified CF/epoxy composites':

'The IFSS is suggested to be dependent on the shape of the fibre. Typically, in perfect adhesion, the IFSS increases non-linearly from the centre of the fibre to a peak value at the end. This is assuming a uniform cylindrical fibre shape during elastic load transfer. However, fibres such as collagen fibrils of the extracellular matrix of connective tissues and nanoparticles such as hydroxyapatite (REF 1) have symmetric tapered ends. During elastic loading, the unique shape of these fibres and particles results in the generation of interfacial shear stresses which are nearly uniformly distributed along the fibre axis. The uniform interfacial shear stress profile serves to minimize the yielding of interfacial bonds. This could otherwise reduce the effectiveness of stress transfer from the matrix to the fibre (or particle) during elastic loading and during fibre pull-out (REF 2). Carbon fibres do not have tapered ends, which makes generating uniform interfacial shear stress along the carbon fibre axis unfeasible. Therefore, modifying the carbon fibre surface using adhesion argument is the best way to address the issue of effective stress transfer. However, the strategies for modifying the carbon fibre surface were mainly aimed at improving the interfacial bond strength. This usually degrades the toughness of the composites and limits their range of applications. Therefore, there is an urgent need for a modification method that can strengthen and toughen the composites at the same time.'

REF 1: Biomedical Materials, 3, 025014, 2008, DOI: 10.1088/1748-6041/3/2/025014

REF 2: Journal of Materials Science, 45,1086–1090, 2010, DOI: 10.1007/s10853-009-4050-2

Question 6

To balance the discussion on the issue of interfacial defects, for the benefit of the readers of this journal, please add the following sentences after Line 233, after ' With the grafting of the coupling agent containing amine functional groups onto the carbon fiber surface, the surface energy of the carbon fiber increases, and allows the matrix resin to better infiltrate the fiber surface and avoiding the appearance of interfacial defects during the preparation of the composite.', i.e. before 'At the same time, the amidated CF is able to bond...':

'When considering the issue of interfacial defects on a smaller length scale, at the level of the CNT, it has been demonstrated that Stone-Wales defects can arise on CNTs (REF 3). This structural imperfection occurs when two pairs of hexagons are transformed into a pentagon-heptagon formation through Stone-Wales bond rotation. The strength of the nanotube is thought to be reliant on various factors, including the nanotube's structure (diameter and chirality) and external conditions such as temperature and strain rate (REF 3). If structural defects are present on the outer shell of MWCNTs, the strength could be notably lower than in the absence of defects (REF 3). However, molecular mechanics predictions indicate that the defects could potentially enhance the mechanical integrity of the CNT by way of defect-defect interactions. It was discovered that at short inter-defect distances, the interatomic bond energy is high, but as the distance between defects grows, the bond energy decreases and stabilizes at a consistent level. High interatomic bond energies are advantageous as they signify that more energy is required to separate the atoms at the defect region, which otherwise present an energically unstable conformation.'

REF 3: Science, 287, 5453, 2000, DOI: 10.1126/science.287.5453.637

No comments

Reviewer 4 Report

Review: Manuscript Materials-2360133

Title: Interfacial enhancement by CNTs grafting towards high-performance mechanical properties of carbon fiber reinforced epoxy composites

The article focuses on the investigation of different methods of improving the interfacial properties of the composite materials and implicitly of the mechanical and hardness properties of the entire composite, by creating a transition layer between the epoxy resin and the carbon nanotubes, using a multiscale reinforcement.

In general, the paper is well structured and organized, but some additions and corrections are still needed:

Introduction

·     1) on page 2, paragraph 65, the sentence "Therefore, a more effective method to immobilize CNTs" must be completed.

Experimental method

·        1) In the section on Materials and Preparation, it is not mentioned how many layers the prepared composite has, how the layers of Carbon fiber fabric were positioned relative to each other, how many samples for tensile testing were prepared for each type of composite. An initial table of the types of samples introduced in section 2.2 would facilitate the understanding of the article.

·        2) What mechanical parameter was measured to establish the improvement of the interfacial properties of the samples, considering that the objective of the study, as stated by the authors, was to find a method to use CNTs to improve the interfacial bonding properties?

·        3) The mechanical test is not enough in this sense. The 3-point bending test could be introduced because during the tests, tangential stresses appear that lead to the appearance of delamination of the layers.

·        4) Also, the hardness was not determined, although it is mentioned that the fact that the new method would lead to its improvement.

Results and discussions

·        1) I recommend introducing a table that centralizes the mechanical and chemical results obtained for the three types of studied samples, highlighting the percentage differences.

 I will recommend for publication the revised manuscript, only after solving/correcting the different important aspects mentioned above.

Reviewer 5 Report

The manuscript “Interfacial enhancement by CNTs grafting towards high-performance mechanical properties of carbon fiber reinforced epoxy composites” addresses an actual problem related to increasing interfacial interaction between the carbon fiber (CF) and the epoxy matrix. For doing, carbon nanotubes (CNTs) were grafted onto the CF surface using the two commercially available coupling agents. It made it possible to significantly improve the roughness and chemical activity of the CF surface. The interfacial interaction was improved through the introduction of a transition layer between the carbon fibers and the epoxy resin. The modified composites showed increased tensile strength, Young's modulus and elongation at break.

The manuscript falls within the scope of the journal of Materials.

The state of the art is well reviewed, with the number of cited papers equal to [26].

The experimental procedure is not described in due details.

The experimental results are given together with their discussion.

The manuscript is summarized with a Conclusion.

The manuscript requires minor revision. The following aspects are to be addressed by the authors.

-          Page 3, line 81. The term “physicochemical structure” does not look as a widespread one.

-          Since the text of the manuscript is rather brief, the information from the supplementary is recommended to be moved to the main text.

-          Page 3, line 88. The abbreviation “AR” is used twice. What does it mean?

-          Page 3, line 94. “CF were modified at the molecular level with KH792 solution”. It was not a molecular modification; this was a surface treatment.

-          Page 4. Lines 123-124. The information on the dimension of epoxy-CF-composite workpieces is not provided. The details on preparing tensile test samples is not given.

-          Page 4, line 137. What does «shous» mean?

-          Page 6, line 191 “area of C-O (286.4 eV), O-C=O (288.6 eV), C-Si (283.9 eV) have”. It should be «has».

-          Page 6, line 207 “and smoothgraphite surfaceshowed”. Please correct the typing.

-          Page 7, line 227. “The mechanical properties of composites with different strengthening routes”. It was not strengthening routes, but CFs’ surface modification ones.

-          Page 7, lines 238-239. “epoxy-functionalized CNTs synergistically constructed a multiscale structure on the CF surface”. The term “synergistically” is often used to stress the improvement of properties, while it is difficult to understand the meaning of “the synergistically constructed a multiscale structure”.

-          Page 7, line 244, 245. “The elongation at break also increased significantly by 35.89%”. Authors tend to provide the highest accuracy in estimating variation of properties. The percentage should be rounded to integral values (this is related to page 7, line 237 as well).

-          Page 9, line 272. “physical anchor riveting effect”. This effect was not mentioned or discussed prior to the Conclusion section.

-          Page 8, line 268. There is a certain disagreement between the manuscript title “Interfacial enhancement by CNTs grafting…” and the first line of conclusion “CNTs-4-KH560-KH792-6-CF multi-scale reinforcement”. It is recommended to harmonized them. 

Minor editing of English language is required.

Round 2

Reviewer 1 Report

The paper can be accepted in its present form.

 Minor editing of English language required.

Reviewer 2 Report

All my comments have been addressed!

Reviewer 4 Report

The paper was improved according to the recommendations made. As a result, I propose its publication in a revised form.